# A reconfigurable arbitrary retarder array as complex structured matter

Chao He [1,13] ✉, Binguo Chen[2,13], Zipei Song [1,13], Zimo Zhao [1,13], Yifei Ma [1,13], Honghui He [2] ✉, Lin Luo [3] ✉, Tade Marozsak [1], An Aloysius Wang [1], Rui Xu[3], Peixiang Huang[3], Jiawen Li[2], Xuke Qiu[1], Yunqi Zhang[1], Bangshan Sun[1], Jiahe Cui [1], Yuxi Cai[1], Yun Zhang[4], Andong Wang[1], Mohan Wang [1], Patrick Salter [1], Julian AJ Fells [1], Ben Dai [5], Shaoxiong Liu[6], Limei Guo[7], Yonghong He[2], Hui Ma [2], Daniel J. Royston [8,9], Steve J. Elston[1], Qiwen Zhan [10], Chengwei Qiu [11], Stephen M. Morris [1], Martin J. Booth [1] & Andrew Forbes [12]

Tuneable retarder arrays, such as spatially patterned liquid crystal devices, have given rise to impressive photonic functionality, fuelling diverse applications ranging from microscopy and holography to encryption and communications. Presently these solutions are limited by the controllable degrees of freedom of structured matter, hindering applications that demand photonic systems with high flexibility and reconfigurable topologies. Here we demonstrate a compound modulator that implements a synthetic tuneable arbitrary retarder array as virtual pixels derived by cascading low functionality tuneable devices, realising full dynamic control of its arbitrary elliptical axis geometry, retardance value, and induced phase. Our approach offers unprecedented functionality that is user-defined and possesses high flexibility, allowing our modulator to act as a new beam generator, analyser, and corrector, opening an exciting path to tuneable topologies of light and matter.

The use of retarders in optics is not new, with waveplates cut from birefringent crystals becoming commercially available as early as the mid-20th century. These homogeneous retarders found use in many optical experiments and soon developed tunability through a variety of different electrically and mechanically driven phenomena. Such components, including photoelastic modulators and liquid crystal devices, are commonplace in optical labs today. In modern nomenclature, retarders are named according to the shape of their fast and slow axes, with linear retarders being a special case of the more general elliptical (arbitrary) case[1,2]. In this work we ask the question: is it possible to construct arbitrary retarder devices that are tuneable, thereby creating a reconfigurable form of structured matter?

[1]Department of Engineering Science, University of Oxford, Parks Road, Oxford OX1 3PJ, UK. [2]Institute of Biopharmaceutical and Health Engineering, Tsinghua Shenzhen International Graduate School, Tsinghua University, Shenzhen 518055, China. [3]College of Engineering, Peking University, Beijing 100871, China. [4]Key Laboratory of Archaeological Sciences and Cultural Heritage, Chinese Academy of Social Sciences, Beijing 102488, China. [5]Department of Statistics, The Chinese University of Hong Kong, Shatin, HK SAR, China. [6]Shenzhen Sixth People's Hospital, Huazhong University of Science and Technology Union Shenzhen Hospital, 518052 Shenzhen, China. [7]Department of Pathology, School of Basic Medical Science, Peking University Health Science Center, Peking University Third Hospital, Beijing, China. [8]Nuffield Division of Clinical Laboratory Sciences, Radcliffe Department of Medicine, University of Oxford, Oxford, UK. [9]Department of Pathology, Oxford University Hospitals NHS Foundation Trust, Oxford, UK. [10]School of Optical-Electrical and Computer Engineering, University of Shanghai for Science and Technology, Shanghai 200093, China. [11]Department of Electrical and Computer Engineering, National University of Singapore, Singapore 117583, Singapore. [12]School of Physics, University of the Witwatersrand, Private Bag 3, Johannesburg 2050, South Africa. [13]These authors contributed equally: Chao He, Binguo Chen, Zipei Song, Zimo Zhao, Yifei Ma. ✉e-mail: chao.he@eng.ox.ac.uk; he.honghui@sz.tsinghua.edu.cn; luol@pku.edu.cn

We find that *tuneable linear* retarder arrays, such as spatial light modulators (SLMs), have a long history, the earliest of which can be traced back to Hamamatsu in the 1980s. There are many ways of realising beam modulators based on *tuneable linear retarder arrays*[3–10], with the dynamic control of metasurfaces[11] being a particularly promising one. This nascent field has already found use in a large range of applications including nanophotonics[12], holography[13] and ultracompact polarimetry[14], with the potential for on-chip integration. In parallel, advancements in fabrication techniques such as photolithography have enabled the construction of *passive arbitrary* retarder arrays, with each point on the array having a designed axis geometry and retardance value (see Ref. 1 for a mathematical description of these parameters). Some techniques that achieve this include the spatial patterning of birefringent substrates[15], selective polymerisation of liquid crystals[16], and more recently, through metasurfaces[13,17,18].

While the linear retarder based devices offer tunability for phase and polarisation control, their use of linear retardation limits the controllable degrees of freedom for manipulating light[3–10], elliptical retarder devices can do arbitrary retardation, but suffer from lack of tuneability. A tuneable arbitrary retarder array could address both challenges simultaneously and has been proposed theoretically[19–21]. However, to date, no experimental realization has been reported.

Here, for the first time, we experimentally realise a tuneable arbitrary retarder array (see Fig. 1a) via virtual pixels with full control over all degrees of freedom by cascading a series of low functionality devices (see Fig. 1b), implemented in our experiments as three SLMs and a deformable mirror (DM). We show that a cascade of elements with judicious encoding can mimic a pixel-controllable arbitrary retarder of any axis geometry, retardance value, and induced phase, simultaneously. This allows us to demonstrate three key applications of this new virtual device (see Fig. 1c). First, we show its power as a reconfigurable, sophisticated beam generator to create complex structured light, including for the first time realising a paraxial optical skyrmion bag, with a random input field. Second, we demonstrate the capability of this modulator as a novel tuneable beam analyser by virtue of its ability to analyse polarisation states simultaneously and provide complete analysing channels, achieving flexible sensing of polarisation sensitive samples. This includes archaeological details which have not been revealed before. Lastly, using the array's direct polarisation aberration compensation capability as a vectorial adaptive corrector, we dynamically correct spatially varying arbitrary retardance aberrations. This approach, employing novel adaptive optics methods, is validated through focus correction in a complex aberrated system.

## Results
### Concept
At the heart of our approach is the notion of virtual pixels through a cascade of lower functionality devices (such as four commercially available, tuneable devices in this work; see Supplementary Method 1) that are used as a tuneable arbitrary retarder array prototype (see Fig. 1b). Note that for all retarders, if and only if such an arbitrary retarder device is mimicked, arbitrary-to-arbitrary state of polarisation (SoP) and phase conversion can be achieved through two conjugated planes in a pixelated manner. This distinguishes our solution from existing cascaded SLM-based methods[3–10], which cannot do arbitrary-to-arbitrary conversion. Importantly, the applications demonstrated in this paper are not limited to our proposed SLM- and DM-based devices; they can be implemented with any device capable of functioning as a tuneable arbitrary retarder array.

Application 1: Beam generator. To demonstrate the broad scope of methodologies and applications that the tuneable arbitrary retarder array can enable, we first utilise it as a new skyrmionic beam generator. Optical skyrmions are topologically protected quasiparticles that have

been recently introduced to the field of structured light[23,24]. However, methods of generating optical skyrmions are still in their infancy, with the common amplitude and phase paradigm achieving low fidelity and being unable to synthesize complex optical skyrmions. Moreover, the limitation of a fixed input uniform SoP restricts flexibility. To address these challenges, we demonstrate a new light manipulation paradigm using our proposed array, based on a light-matter interaction involving arbitrary retarder. This enables, for the first time, the generation of paraxial beams with complex topologies such as optical skyrmion lattices and bag.

The procedure of beam generation is as follows: First, an arbitrary field with known polarisation and phase is incident upon the array (here we use a beam with circular polarisation and a flat phase for simplicity). Second, we determine the expected output fields. Third, we calculate the corresponding spatial retarder distributions and encode them into the retarder array sequentially (see Supplementary Method 1 and 2). Fourth, we use a Stokes polarimeter to record the beam profiles[25]. We then verified the feasibility of generating complex beams. As shown in Fig. 2a, the theoretical and experimental results of six representative profiles confirm that the complex skyrmionic beams generated by our array well-matched their corresponding ground truths. We also show two examples of the measured Mueller matrices (see Fig. 2b) of the array, obtained using the conventional dual rotating waveplate polarimetry technique[26], alongside the corresponding theoretical profiles, to demonstrate their consistency with our expectations in the object's point of view. We then further tested a random non-uniform input field (see Fig. 2c) through the same procedure as before, and found that the device is able to convert the field into optical skyrmion bag, thereby validating that it circumvents the need to pre-determine the input SoP and phase. Two examples are shown in Fig. 2d to illustrate the wrapped relationship between these complex optical fields (unit skyrmion and bag) and the 2-spheres, representing a form of stereographic projection.

Note any paraxial topological structured field that exists in the plane perpendicular to the direction of propagation can theoretically be realised by our array through the same process. This potentially includes exotic states that have both non-trivial orbital angular momentum and non-trivial skyrmion number. Furthermore, the utility of such tuneable retarder arrays for dynamic beam generation allows us to easily investigate the properties of different topologically structured fields under the same platform and experimental condition. As an example, we show topological protection for two beams through uniform isotropic and anisotropic perturbations (see Supplementary Method 3 and Supplementary Note 2).

Application 2: Beam analyser. The arbitrary retarder array can serve as tuneable polarisation analysis channels. Full Poincaré units (FPU)[27], taking advantage of their unique properties in the analysis of all SoPs simultaneously, have been used to achieve high-precision polarisation measurement through physical inference, circumventing the error accumulation problem of traditional measurement processes[27]. However, the realisation of tuneable polarisation imaging using FPUs is limited by existing devices, as they are not optimised for retarder distribution or passivity[27]. Consequently, they cannot meet the requirements of applications that demand distinct optimisation strategies for SoP sensing. Our array overcomes these limitations by enabling the formation of different FPUs with optimised states (for circular SoPs or linear SoPs) through controllable retarder distributions.

The procedure proceeded as follows: first, to realise optimised FPUs that are designed for circular SoPs (new method 1) or linear SoPs (new method 2) optimisation, we encode target spatially varying linear retarder distributions into the array (details in Supplementary Method 4). We then measured the Mueller matrices of the matter in both configurations to ensure they are of expected formation (see Supplementary Note 3); second, to validate the feasibility of the FPU

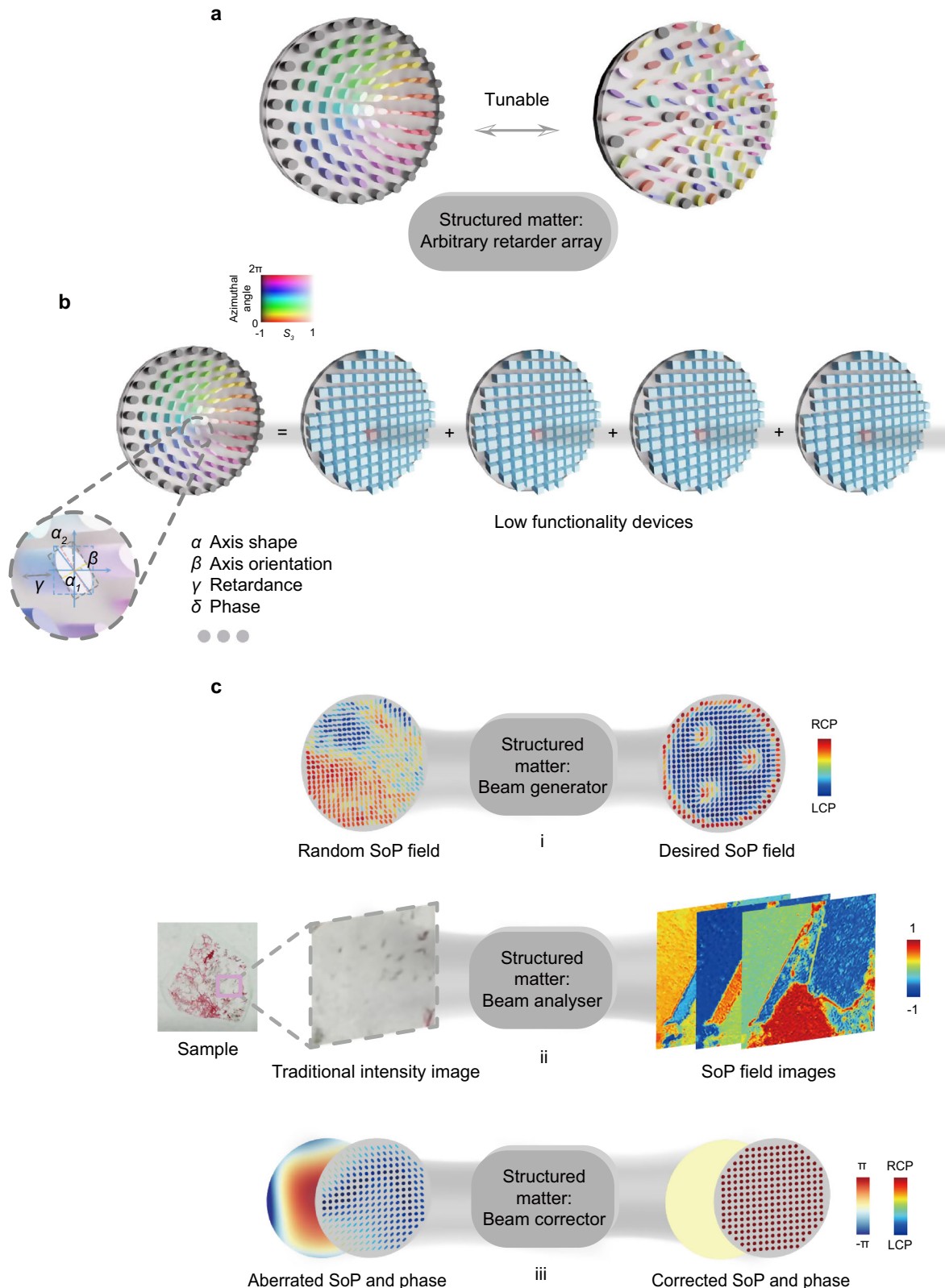

**Nature Communications** | (2025)16:4902

device, a polarisation scanning system was adopted, where the polarisation state analyser is formed by the array and a fixed polariser. This configuration enables proof-of-concept experiments on the characterisation of target samples via an optimised FPU array imaging system (see Supplementary Note 3); third, we used a circular SoP illumination and acquired polarimetric images of an archaeological and a biomedical sample. We affirm that samples were collected in a

responsible manner and in accordance with relevant permits and local laws. The archaeological sample is a red pigment originating from cinnabar mines in China during the Qin and Han dynasties, where the fine birefringent microstructure of cinnabar is crucial for research into tracing the mineral's provenance. The biomedical sample is a bone marrow trephine from a myelofibrosis case, where spatial retardance reveals the accumulation of fibrotic structures—critical for

**Fig. 1 | Tuneable arbitrary retarder array and the applications. a** A schematic of a tuneable elliptical (arbitrary) retarder array. The shape and colour of each pillar indicate different retarder axis geometries, while the pillar height corresponds to its retardance value. A vector representation (like the Stokes vector using $S_1$, $S_2$, and $S_3$) is used to describe the axis geometry. For visualization, we use hue to represent the azimuthal angle $\tan\theta = \frac{S_2}{S_1}$ and lightness to indicate the magnitude of $S_3$, similar to ref. 22,23. **b** A tuneable arbitrary retarder array-based device can be formed via reconfigurable pixels, allowing control over axis geometry $\alpha$, orientation $\beta$, retardance value $\gamma$, and induced phase $\delta$, allowing its functionality to be switched on demand. However, these devices do not physically exist. Through this work, the construction of the device is achieved by four cascading low-functionality devices

such as SLMs and DM (see Supplementary Method 1), hence forming virtual pixels that together act as a synthetic form of reconfigurable structured matter. Note that for all applications conducted in this paper, a precise calibration of such an array must first be performed (see Supplementary Method 2 and Supplementary Note 1, which includes device performance assessment) to achieve accurate pixelated control of arbitrary-to-arbitrary SoP and phase conversion. **c** Three functions that the arbitrary retarder array can perform: a novel beam generator, beam analyser, and beam corrector. The polarisation ellipses are plotted and coloured according to the magnitude of the last component of the Stokes vector to illustrate the transition from left circular polarisation (LCP) to right circular polarisation (RCP).

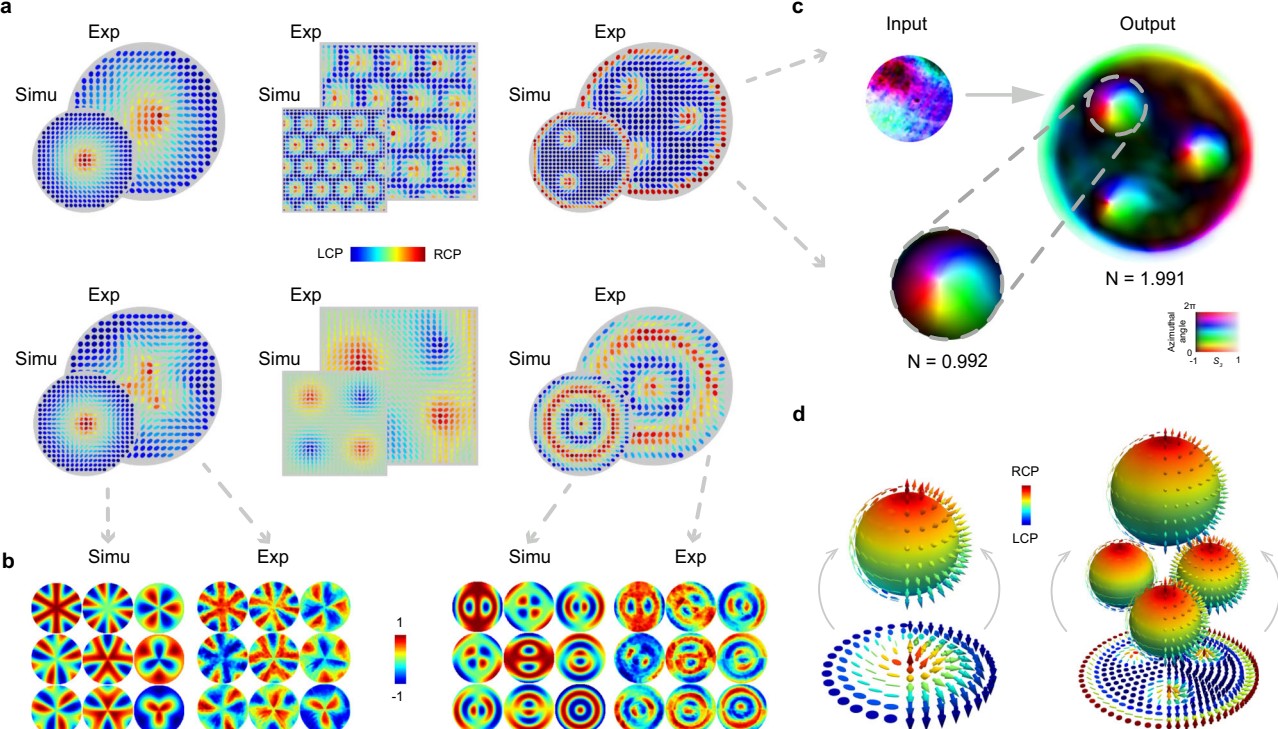

**Fig. 2 | Tuneable arbitrary retarder array as a new beam generator. a** Six generated complex skyrmionic beams with both simulated and experimental results, including: individual optical skyrmions, optical skyrmion lattices, meron lattices, high-order optical skyrmions with azimuthal and radial variations, and skyrmion bag. **b** Theory and experimental results of the Mueller matrices of the two high-order optical skyrmion generators. **c** The optical skyrmion bag generated with a

non-uniform input beam. The skyrmion number, as well as the unit skyrmion inside the bag has been demonstrated. **d** Visualization of optical skyrmion and skyrmion bag mapped onto the Poincaré and Bloch spheres, illustrating their topological properties. The vectorial arrows in space (as well as the Stokes ellipses in space representing SoP), which can be stereographically mapped from a Bloch sphere (as well as Poincaré sphere) that holds the skyrmionic state.

pathological analysis, subsequent clinical staging, and precise treatment planning. In both applications, our approach adds tremendous value as an analyser, with the results shown in Fig. 3a, b. Both figures share a similar format, including a real-world sample, a traditional intensity image, a decomposed polarimetric image[28], as well as a quantitative data comparison of two selected zones (Z1 and Z2). It shows that compared with traditional method, our new methods 1 and 2 are both capable of distinguishing information within the samples, while maintaining a high level of agreement with the ground truth. See Supplementary Note 3 for more details.

Note that this is the first time a tuneable FPU-based imaging platform has been realized, paving the way for future dynamic sensing and tailored optimization strategies across a wide range of target objects.

Application 3: Beam corrector. Besides being a novel beam generator and beam analyser, arbitrary retarder arrays can act as a novel dynamic aberration corrector for adaptive optics (AO). To meet the high demand of rapidly developing vectorial optics and photonics, AO

techniques that address both polarisation and phase aberration correction have been recently introduced—named vectorial adaptive optics[29]. However, latest methods still cannot perform arbitrary SoP and phase conversion, which is essential for key applications such as aberration correction in the detection pathway[29–31]. Additionally, they are unable to eliminate the complex field conversion process required for sensorless correction procedures[29]. One possible approach is to develop an arbitrary SoP and phase converter, making the vectorial AO corrector universal and suitable for any location within the optical system, while effectively circumventing the need for field conversion. However, to the best of our knowledge, this is not readily achievable using existing devices. Theoretical work has just been recently proposed[29–31]. In this work, for the first time, we realise a new vectorial AO corrector using our array by harnessing its arbitrary SoP and phase conversion capabilities. This approach introduces two new AO methodologies—sensor-based and sensorless AO—by effectively performing retarder compensation from the object's perspective, hence termed object-wise AO (O-AO). As a proof of concept, we demonstrate the

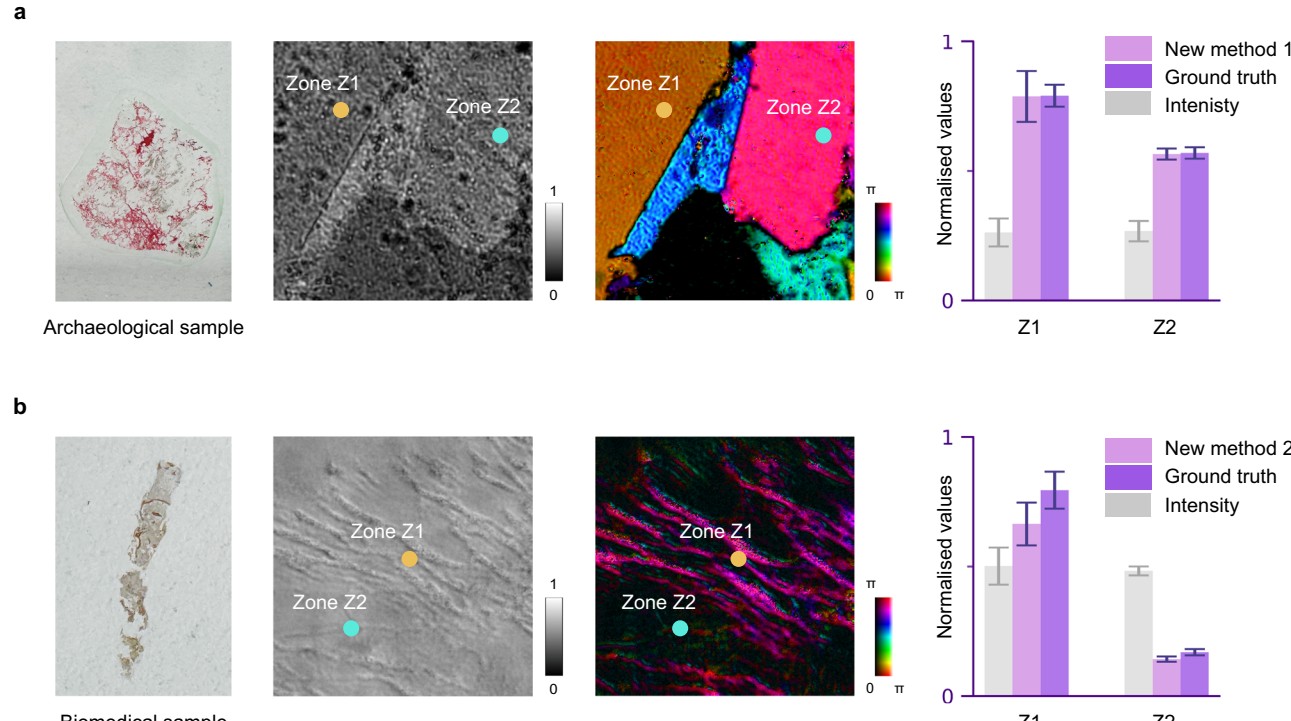

**Fig. 3 | Tuneable arbitrary retarder array as a new beam analyser. a** An archaeological sample exhibiting intrinsic birefringence. The real-world picture of the slice, its intensity image under a traditional microscope, and the corresponding polarimetric image are given. Zones Z1 and Z2 represent two small chosen regions for quantitative analysis. The statistical comparison of normalized values of the intensity, decomposed polarimetric parameter, and ground truth taken by traditional polarisation microscope are given, with data obtained from Z1 and Z2. In the HSL colour map used, the hue (H) represents the fast axis orientation; the lightness (L) reflects the retardance of the measured area; and the saturation (S) is the constant 1. **b** A biomedical sample featuring fibrosis properties. The real-world picture of the slice, its intensity image under a traditional microscope, and the corresponding polarimetric image are given. A similar statistical analysis has been also conducted to form the bar chart.

feasibility of both methods for correction in the detection path of a focusing system (see Supplementary Note 4).

Experimental procedures for validation of sensor-based O-AO are as follows: First, an external polarisation aberration was introduced into the system (a tilted waveplate array[29]). Second, a Mueller matrix polarimeter was used to measure the existing aberration. Third, a calculated compensation retarder was formed using the corrector to finish the compensation. Detailed procedures are described in Supplementary Method 5 and Supplementary Note 4. Fourth, we ran sensorless phase AO correction to compensate for the residual phase errors. The measured Mueller matrix for aberration, the compensation pattens on four devices (i.e., the corrector), as well as a cross-section comparison of the focal spots (O-AO off vs. on) are shown in Fig. 4a to validate the improvement.

Validation of sensorless O-AO (without a Mueller matrix sensor) was also conducted, by optimising the focus intensity to find the best compensation. Note we employ a newly defined set of object-specific modes, similar to Zernike modes, which we call retardance modes (see Supplementary Method 5) to estimate polarisation aberration. Experimental procedures for validation of its effectiveness are as follows: First, we applied different retardance modes on our array. These modes were used to manipulate the retardance distribution on the array in a pixelated manner with varying parameters, and focal spot images were recorded sequentially (see details in Supplementary Method 5 and the flowchart in Supplementary Note 4). Second, the recorded images provided information that allowed for the estimation of retardance aberrations. Third, the estimated aberrations were used to drive the array to correct the aberration (see Supplementary Method 5). Figure 4b presents phase pattern and focal spot analyses

akin to those described in the sensor-based section, with the addition of a fitted curve for a retardance mode—piston, which is usually neglected in traditional AO. Note in this case, the piston mode represents the object's retardance value, which alters the overall retarder performance and, in turn, affects the focal spot. It is evident that our O-AO successfully recovered the focal spot.

To the best of our knowledge, this is the first experimental demonstration of a universal vectorial AO corrector enabled by array driven arbitrary vectorial field conversion, achieving capabilities unattainable by previous devices. This advance has great potential to benefit applications from materials characterization to clinical diagnostics.

## Discussion

Our results show that a tuneable arbitrary retarder array can enable the realisation of various downstream optical techniques and applications. Proof-of-concept demonstrations in this work validate how the array can be effectively used as a beam generator, analyser, and corrector.

We note again that while cascaded SLMs have indeed been utilised previously[3–10], *none* of them possess the capability to construct a reconfigurable arbitrary elliptical retarder array. It is only through the realisation of the proposed array, with simultaneous control over its axis geometry, orientation, retardance value, and induced phase (note this phase reflects the properties of the object—such as its refractive index or geometric structure—rather than the phase of the light itself), can arbitrarily SoP and phase conversions be enabled. This advancement unlocks a range of new possibilities, as discussed earlier and later in the text.

As we mentioned before, for all three applications, we conduct a systematic calibration of the four devices, bringing them to an

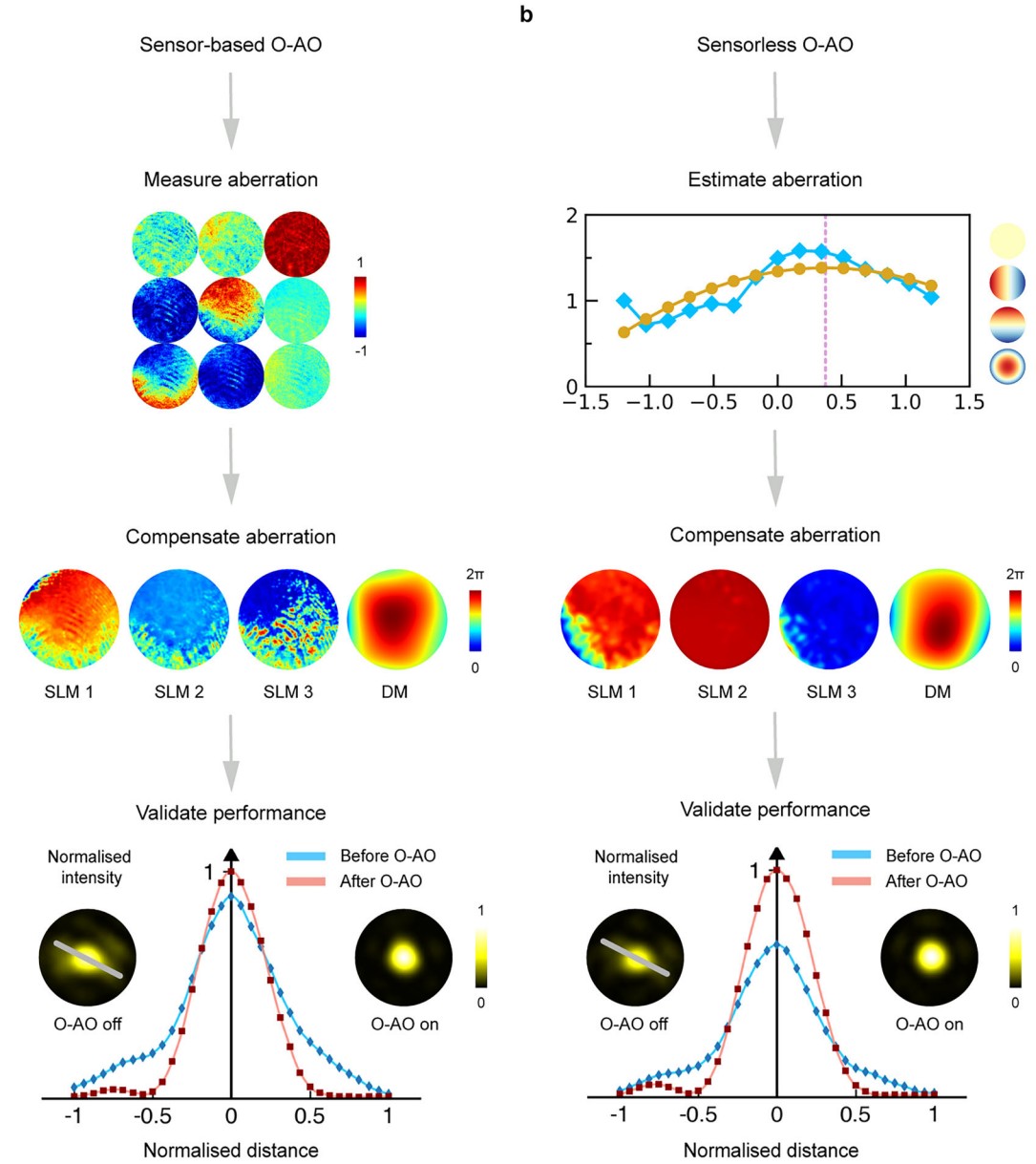

**Fig. 4 | Tuneable arbitrary retarder array as a new beam corrector. a** Results for sensor-based O-AO off vs. on. Measured Mueller matrix of the aberration, phase patterns on all four devices after O-AO correction, foci profiles as well as sampled focal spot cross-sections (white line on the focal spot image) before and after correction are presented. The vectorial aberration is a tilted waveplate array[29]. Y-axis in the line chart represents the normalized intensity value and x-axis represents the distance of the sampled curve on the focal spot. **b** Results for sensorless O-AO off vs. on for the same retardance aberration in (**a**). A fitted curve for retardance piston mode is given as an example for aberration estimation. Phase patterns on all four devices after O-AO correction, foci profiles as well as sampled focal spot cross-sections before and after correction are also presented like (**a**).

optimised status for subsequent beam control (see Supplementary Methods 1 and 2, and Supplementary Note 1). Once the devices are ready, the SLM and DM patterns are calculated by solving the inverse problem between the known input states and the target vectorial beams. The corresponding phase distributions for the four devices are derived and encoded using a calibration-based look-up table, as explained in Supplementary Method 2. This optimisation framework ensures that the array functions across different applications with high precision and flexibility. Generator: utilising our array with a pixel-by-pixel plane mapping mechanism, complex skyrmionic beams with arbitrarily intricate topologies can be generated in a pixelated, tuneable manner, without being restricted by input states. This modulator offers high flexibility for structured light generation in practical applications and opens new avenues for research in next-generation dynamic optical communication. Analyser: the array provides flexible

analysis channels, tailored to specific applications. This flexibility paves the way for novel optimisation geometries, such as elliptical SoP configurations or hybrid states customised for individual needs. Corrector: the array (i.e., the new AO corrector) expands the modern vectorial AO toolbox, enabling the correction of polarisation aberrations that previous devices could not address. Future research could explore more advanced vectorial modes, such as axis geometry manipulation. These implementations may benefit research areas beyond those discussed in this paper, including direct laser writing and lithography technology.

Our proposed implementation of a tuneable arbitrary retarder array does, however, have drawbacks related to the use of cascaded devices (SLMs and DM), such as the modulation efficiency, mass and size, alignment and resolution. However, these don't penalise the general applicability of tuneable arbitrary elliptical retarder arrays but

are rather considerations due to the limitations in current technology. In the future, we expect these drawbacks to be overcome by next generation devices such as those made by liquid crystal based meta-surfaces and metamaterials[11], or phase-changing materials[12,32]—as theoretically there are numerous ways to realize such arbitrary retarder arrays. While this work has focused on three new applications of the compound modulator in the format of arbitrary retarder arrays, there exists other complex structured matter such as tuneable diattenuator arrays or depolariser arrays (see Supplementary Methods 6, 7, 8, and Supplementary Note 5; can also be compounded by cascading devices in a synthetic manner), which further offer pixelated control of intensity and degree of polarisation[33], to open various new opportunities.

Overall, we expect arbitrary and tuneable retarder arrays to play an important role in the modern optics community, especially in the fields of dynamic and vectorial optics and photonics. We believe that this development can address recent demands and open the door to more widespread applications of such tuneable arrays, and in fields ranging from beam steering[34] to optical communications[35], polarimetry[25] to sensing[36], adaptive optics[21], optical skyrmions, and beyond.

## Data availability

All the main data supporting the results of this study are available within the paper, Supplementary Information and Source data. Supplementary Information and Source data for all figures are provided in this paper. The data that supports the plots within this paper and other findings of this study are available from the corresponding author upon request. Source data are provided with this paper.

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

## Acknowledgements

We would like to thank the support of St John's College, the University of Oxford, and the Royal Society (URF\R1\241734) (C.H.); the Engineering and Physical Sciences Research Council (UK) (EP/R004803/01) (P.S.S.); the European Research Council (AdOMiS, no. 695140) (C.H. and M.J.B.); NSFC 92050202 (Q.Z.); Shenzhen Key Fundamental Research Project (No. JCYJ20210324120012035) (H.H.).

## Author contributions

B.C., Z.S., Z.Z., Y.M. and C.H. conceived the main ideas, developed the concepts, and planned the simulation and experiments. Z.Z., T.M. and A.A.W. performed mathematical approaches, Z.S., C.H. and B.C.

contributed to the simulation work. Y.M., C.H., YQ.Z. and Z.Z. performed experimental validation. X.Q., YQ.Z., B.S. and Y.C. analysed the experimental results of skyrmionic beam section. R.X., P.H., B.D., J.L., Y.H., D.J.R., H.M., S.L., L.G. and Y.Z. analysed the experimental results of polarimetric imaging section. J.C., A.W., J.A.F., and M.W. analysed the experimental results of adaptive optics section. Z.S., P.S., B.D., Q.Z., C.Q., S.J.E., S.M.M., M.J.B., A.F., and C.H. analysed the experimental results of the SLM cascades and provided instructions for applications. Y.M., Z.Z., C.H. and A.F. prepared the figures. Z.Z., Y.M., C.H., and A.F. wrote and reviewed the paper. H.H., L.L., C.H., and A.F. provided supervision. H.H. and L.L. led the separated parts of this project and C.H. led the overall project. All authors reviewed the results and approved the final version of the manuscript.

## Competing interests

The authors declare no competing interests.
