## [Transparent Peer Review file · Nature Communications]

A reconfigurable arbitrary retarder array as complex structured matter

Corresponding Author: Dr Chao He

Version 0:

Reviewer comments:

Reviewer #1

(Remarks to the Author)

I read carefully the authors' response to my comments, and I'm generally satisfied with the corrections they made to their manuscript. I think the novelty of their device and its performances are much clearer described, making it now possible to appreciate the meaningfulness of their work. I should acknowledge as well that they made a good job in cleaning up the text and figures, making the whole manuscript much easier to read.

My only remaining concern is linked to application 2 where they study real-life samples (biological and archeological). I still can't appreciate the soundness nor the meaningfulness of those applications. These are completely out of my field and rather far from the general framework of the manuscript. I understand they want to highlight the performance of their device with real-life applications. However, these are so far away from my expertise (and I would think of that of people interested in such work) that this is not useful to someone like me who simply wants to understand the basic principles of such a device. Furthermore, the right-most part of Fig. 3 is very cryptic to me: there are no axis labels and I can't interpret these data appropriately. I simply can't review this part, and I would advise (as I did before) to simply remove this part and prepare a separated publication in a journal specialized on such topic.

I can recommend the manuscript as it is right now, but I would strongly advise to remove Application 2.

Reviewer #2

(Remarks to the Author)

I would like to commend the authors for their substantial efforts in revising the manuscript, which has resulted in significant improvements. The enhanced clarity in the motivation and novelty statements, along with the addition of new experimental results demonstrating the advantages of structured vector beams for the presented applications, has greatly strengthened the work. In my opinion, the manuscript, in its current form, is now well-suited for publication in Nature Communications.

Reviewer #3

(Remarks to the Author)

The authors experimentally demonstrate the use of 3 SLMs and 1 DMD to generate an arbitrary phase retarder and polarization control over a laser beam.

Following the theoretical model reported by some of the authors in [Ref. 21, Hu Q, He C, Booth MJ. Arbitrary complex retarders using a sequence of spatial light modulators as the basis for adaptive polarization compensation. Journal of Optics. 2021;23(6):065602.], they show how it is possible to apply this optical manipulation for 1) generate skyrmion beams, 2) analyze the polarization of a beam in combination with a fixed polarizer, and 3) correct for undesired polarization states.

From the technological point of view, the use of cascaded SLM and DMD is not such a relevant innovation. However, the paper reports a very interesting method of light polarization manipulation [ref. 21] that could interest several readers if it integrates experimental information on the configuration and settings (programming) of SLM and DMD and a more in-depth

discussion of the results. The content of the manuscript, in my opinion, could be better suited to a more specialized journal.

I would suggest the authors to address also the following points:

- The optimization of the SLM phase profile and the DMD pattern is not clear for the different applications. I would suggest to discuss the calculated patterns more generally also in the main text and to give some examples. The reader would also benefit from a clearer description of the symbols and colors used in the images for the Stokes parameters. Are the pixels controlled individually or grouped in areas?
- What are the instruments used in the experiments? The experimental setup and the schemes in the supplementary notes would benefit from a more detailed description of the individual components.
- Figure S1c: Distance units are missing (?). How was the precision defined?
- Application 1: How were the spatial distributions of the retarders calculated in more general terms? Figure 2a: How were the experimental maps retrieved from the measurements made by the Stokes polarimeter? Figure 2b: How were the 9 (experimental) Mueller matrices measured?
- Application 2: Figure 3a: The colored bars of the polarimetric images are not clear, as well as the y-axis of the statistical comparison for the intensities and the polarimetric parameter. A double y-axis could clarify the plot. Regarding the Mueller matrices: how does the polarimetric parameter depend on the number of repeated units (if any) on the SLMs? Figure S3d: It would be interesting to see the two Cartesian and Polar optimization methods applied to the two samples. What are the Stokes parameters measured with the polarimeter (ground truth) compared to those obtained with the proposed methods? Can the authors comment on the advantages of using this method compared to the one reported in Ref. 25?
- Application 3: How many iterations were performed in the O-AO (sensor and sensor-less) approach? How much does the convergence of this process depend on the deviation of the input beam from the desired output? Are there any limitations?

Version 1:

Reviewer comments:

Reviewer #3

(Remarks to the Author)

I appreciate the Authors' detailed responses and revisions to the text. The manuscript now includes more experimental details, better explains the results obtained and the procedure used. This makes the work clearer and more reproducible for the wide readership of Nature Communication. I recommend publication of the article.

General Reply to All Reviewers

We sincerely thank all reviewers for their constructive and positive feedback, which has significantly enhanced the quality and clarity of our manuscript.

Our responses are provided below in blue following the comments raised by each reviewer (shown in italics). We provide a **Point-to-Point Reply to All Comments of Reviewers**, which we hope will be beneficial in clarifying the main changes made to the manuscript and addressing overarching points raised during the review. Revisions to the Main article and Supplementary Notes are highlighted in **Yellow** in the revision files and are also indicated in **Purple** text in this response below.

Reply to Reviewer #1

Comment: *I read carefully the authors' response to my comments, and I'm generally satisfied with the corrections they made to their manuscript. I think the novelty of their device and its performances are much clearer described, making it now possible to appreciate the meaningfulness of their work. I should acknowledge as well that they made a good job in cleaning up the text and figures, making the whole manuscript much easier to read.*

My only remaining concern is linked to application 2 where they study real-life samples (biological and archeological). I still can't appreciate the soundness nor the meaningfulness of those applications. These are completely out of my field and rather far from the general framework of the manuscript. I understand they want to highlight the performance of their device with real-life applications. However, these are so far away from my expertise (and I would think of that of people interested in such work) that this is not useful to someone like me who simply wants to understand the basic principles of such a device. Furthermore, the right-most part of Fig. 3 is very cryptic to me: there are no axis labels and I can't interpret these data appropriately. I simply can't review this part, and I would advise (as I did before) to simply remove this part and prepare a separated publication in a journal specialized on such topic.

I can recommend the manuscript as it is right now, but I would strongly advise to remove Application 2.

Response:

We sincerely thank the reviewer for their positive feedback, support for publication, and for recognising the improvements made in the manuscript. We are also grateful for the insightful comments and suggestions, which have significantly strengthened the clarity and overall quality of our work.

For Application 2, please allow us to keep it and explain the reasons: 1) We very much appreciate the comments and understand that this is more suitable for a targeted community (such as polarisation imaging and sample analysis). As we reported before, we just not only wanted to put forward such interesting devices (arbitrary tuneable retarder array) to demonstrate arbitrary pixelated polarisation and phase conversion (i.e., arbitrary vectorial field conversion), but also to point out their applications -- as a beam generator, beam analyser, and beam corrector -- as a systematic demonstration to potentially attract interest from different optics communities. This might benefit interdisciplinary communication and inspire more potential research directions. 2) However, we still greatly appreciate the reviewer's valuable comments, so we have again revised Application 2 following the reviewer's comments. We have added further clarifications to the Fig. 3 and its caption, as well as a more detailed description of the sample, data and the associated analysis for additional clarity (these revisions have been highlighted in the content related to Application 2, in both Main article and SI Note). We hope the revised version meets the reviewer's expectations. Many thanks indeed again.

Reply to Reviewer #2

Comment: *I would like to commend the authors for their substantial efforts in revising the manuscript, which has resulted in significant improvements. The enhanced clarity in the motivation and novelty statements, along with the addition of new experimental results demonstrating the advantages of structured vector beams for the presented applications, has greatly strengthened the work. In my opinion, the manuscript, in its current form, is now well-suited for publication in Nature Communications.*

Response: We sincerely thank the reviewer for the positive feedback, support for publication and for recognising the improvements made in the manuscript. We are grateful for the insightful comments and suggestions, which have significantly strengthened the clarity and overall quality of the work.

Reply to Reviewer #3

The authors experimentally demonstrate the use of 3 SLMs and 1 DMD to generate an arbitrary phase retarder and polarization control over a laser beam.

Following the theoretical model reported by some of the authors in [Ref. 21, Hu Q, He C, Booth MJ. Arbitrary complex retarders using a sequence of spatial light modulators as the basis for adaptive polarization compensation. Journal of Optics. 2021;23(6):065602.], they show how it is possible to apply this optical manipulation for 1) generate skyrmion beams, 2) analyze the polarization of a beam in combination with a fixed polarizer, and 3) correct for undesired polarization states.

From the technological point of view, the use of cascaded SLM and DMD is not such a relevant innovation. However, the paper reports a very interesting method of light polarization manipulation [ref. 21] that could interest several readers if it integrates experimental information on the configuration and settings (programming) of SLM and DMD and a more in-depth discussion of the results. The content of the manuscript, in my opinion, could be better suited to a more specialized journal.

We are very grateful for the reviewer's positive comments, advice, and recognition of our work. We have tried our best to address the reviewer's comments and made the related modifications throughout the paper (both the Main article and SI Notes). Before we provide a point-by-point reply and outline the detailed changes, please allow us to clarify a few points:

First, we sincerely appreciate and acknowledge that various cascaded SLMs, or SLM-plus-DM approaches, have been proposed before -- such as those cited in the main article (Refs¹⁻⁸) as well as Ref [21], which the reviewer suggested, and we also cited in previous version already. However, aside from the model in Ref [21] (which is not only a theoretical work but also specific to point adaptive optics applications only), **NONE** of the others have ever been used to achieve arbitrary pixelated polarisation and phase to arbitrary pixelated polarisation and phase conversion in a field-based manner (i.e., arbitrary vectorial field conversion), which is exactly what are demonstrated here.

In this work, we not only formally introduce the concept of a tuneable arbitrary elliptical retarder but also, for the **FIRST** time, experimentally create a "virtual device" with remarkable new functionality (as a tuneable arbitrary elliptical retarder array) and applications. This clearly advances the state of the art, as our virtual device is highly versatile and reconfigurable, combining the best features of both artificial atoms (metasurfaces) and reconfigurable liquid crystal devices (SLMs). Moreover, we go further by utilizing this virtual device in a range of exciting applications, from novel anti-perturbation topological beam generation to probing exotic samples, demonstrating its immediate potential for impact across multiple research communities.

Finally, we note that we have included detailed procedures and protocols for all three applications in the Methods section, as well as comprehensive supplementary data and their analysis in the SI Notes. We hope the reviewer is now happy with our revised version.

1. Moreno, I., Davis, J. A., Hernandez, T. M., Cottrell, D. M. & Sand, D. Complete polarization control of light from a liquid crystal spatial light modulator. *Opt Express* 20, 364–376 (2012).
2. Davis, J. A., Evans, G. H. & Moreno, I. Polarization-multiplexed diffractive optical elements with liquid-crystal displays. *Appl Opt* 44, 4049–4052 (2005).
3. Rubin, N. A., Shi, Z. & Capasso, F. Polarization in diffractive optics and metasurfaces. *Adv Opt Photonics* 13, 836–970 (2021).
4. Han, W., Yang, Y., Cheng, W. & Zhan, Q. Vectorial optical field generator for the creation of arbitrarily complex fields. *Opt Express* 21, 20692–20706 (2013).
5. Zhong, R.-Y. et al. Gouy-phase-mediated propagation variations and revivals of transverse structure in vectorially structured light. *Phys Rev A* 103, 053520 (2021).
6. Li, Y. et al. Single-exposure fabrication of tunable Pancharatnam-Berry devices using a dye-doped liquid crystal. *Opt Express* 27, 9054–9060 (2019).
7. Rong, Z.-Y., Han, Y.-J., Wang, S.-Z. & Guo, C.-S. Generation of arbitrary vector beams with cascaded liquid crystal spatial light modulators. *Opt Express* 22, 1636–1644 (2014).
8. Wang, X.-L., Ding, J., Ni, W.-J., Guo, C.-S. & Wang, H.-T. Generation of arbitrary vector beams with a spatial light modulator and a common path interferometric arrangement. *Opt Lett* 32, 3549–3551 (2007).

Comment 1: I would suggest the authors to address also the following points: The optimization of the SLM phase profile and the DMD pattern is not clear for the different applications. I would suggest to discuss the calculated patterns more generally also in the main text and to give some examples. The reader would also benefit from a clearer description of the symbols and colors used in the images for the Stokes parameters. Are the pixels controlled individually or grouped in areas?

Response 1: Many thanks for the valuable comments. Please let us address them one by one below:

1) At the very beginning, we need to clarify that before all applications, we conduct an overall calibration of all SLMs and the DM, bringing them to the same optimal status for later use. The mechanism and calibration process are further detailed in Methods 1 and 2, and in SI Note 1, where we provide the patterns before and after calibration (in Stokes vector ellipses, separated components S_1 , S_2 , S_3 , DM phases). Additionally, we introduce the correctness and uniformity of vectorial information as important criteria for evaluating the optimisation of the device (also attached below the results). Note more details can be found in the replies to the Reviewers' comments 2 and 3 below.

Calibration of the tuneable arbitrary retarder array (details see SI Note 1)

Once the optimised devices are ready, the SLM and DM patterns are calculated based on the specific requirements of different applications by solving the inverse problem between the known input states and the target vectorial beams. The corresponding phase distributions for the four devices are derived and encoded onto the devices using a calibration-based look-up table, as explained in Method 2. The properties of the overall arbitrary retarder arrays are recorded described using Mueller matrices (Fig. 2b for Application 1; Fig. S3 for Application 2) or directly through the four phase patterns (Figs. 4a and 4b) to validate the feasibility again.

We have also modified the Main article (Discussion and Conclusion, paragraph 3) to clarify how the optimisation framework supports the multifunctionality of the array: "As we mentioned before, for all three applications, we conduct a systematic calibration of the four devices, bringing them to an optimised status for subsequent beam control (see Methods 1 and 2, and Supplementary Note 1). Once the devices are ready, the SLM and DM patterns are calculated by solving the inverse problem

between the known input states and the target vectorial beams. The corresponding phase distributions for the four devices are derived and encoded using a calibration-based look-up table, as explained in Method 2. This optimisation framework ensures that the array functions across different applications with high precision and flexibility."

2) We have also provided a clearer description of the symbols and colours used in the images for the Stokes parameters (representing both axis geometry and light properties) by updating the captions in Fig. 1. The caption for Fig. 1a has been revised as follows: "(a) A schematic of a tuneable elliptical (arbitrary) retarder array. The shape and colour of each pillar indicate different retarder axis geometries, while the pillar height corresponds to its retardance value. A vector representation (like the Stokes vector using S_1 , S_2 , and S_3) is used to describe the axis geometry. For visualization, we use hue to represent the azimuthal angle $\tan \theta = \frac{S_2}{S_1}$ and lightness to indicate the magnitude of S_3 , similar to Ref^{9,10}." The caption of Fig. 1b has been updated as: "(b) A tuneable arbitrary retarder array-based device can be formed via reconfigurable pixels, allowing control over axis geometry α , orientation β , retardance value γ , and induced phase δ , ..." Also, the caption of Fig. 1c has been revised as: "(c) ... The polarisation ellipses are plotted and coloured according to the magnitude of the last component of the Stokes vector to illustrate the transition from left circular polarisation (LCP) to right circular polarisation (RCP)."

3) Each pixel on the SLM and DM is controlled independently without grouping, enabling high-resolution modulation and maximum tunability. However, for visualisation purposes, the phase profile on the DM is illustrated and the Stokes ellipses are plotted from down-sampled measurement data using a 10x10 averaging kernel at 10-pixel intervals to improve clarity while preserving polarisation characteristics, ensuring accurate and informative visual representation. The caption of Fig. S1b in SI Note 1 has been revised to address this point: "(b) ... To enhance visual clarity, the phase profile on the DM is illustrated and the Stokes ellipses are plotted based on down-sampled measurement data averaged over neighbouring 10x10 pixels. "

9. Wang, A. A. et al. Topological protection of optical skyrmions through complex media. *Light Sci Appl* **13**, 314 (2024).

10. Shen, Y. et al. Optical skyrmions and other topological quasiparticles of light. *Nat Photonics* **18**, 15–25 (2024).

Comment 2: *What are the instruments used in the experiments? The experimental setup and the schemes in the supplementary notes would benefit from a more detailed description of the individual components.*

Response 2: We thank the reviewer for the insightful comment and for highlighting the need for clarity regarding the experimental setup. We have updated Fig. S1a in SI Note 1 (see Figure in Response 1) to include a more detailed illustration and description of each component in our experimental setup. The caption has also been revised as follows: "(a) Schematic of the experimental setup. The system consists of a He-Ne laser (Melles Griot, 05-LHP-171, 632.8 nm) as the light source, followed by a beam expander (BE). The tuneable arbitrary retarder array is composed of three SLMs (SLM1, SLM2, SLM3; Hamamatsu, X10468-01) and a DM (Boston Micromachines Corporation, Multi-

3.5). Half-wave plates (HWPs; Thorlabs, WPH05M-633) are placed between the SLMs to introduce a 45° relative rotation. A rotating quarter-wave plate (QWP; Thorlabs, WPQ10M-633) and a fixed polarizer (P; Thorlabs, GL10-A) are used for polarimetric measurements, and the output is recorded by a camera (Cam; Thorlabs, DCC3240N).”

We hope these additional details clarify how the instruments are arranged and used in our experiments.

Comment 3: *Figure S1c: Distance units are missing (?). How was the precision defined?*

Response 3: Many thanks for the valuable comments. Please allow us to explain:

1) In Fig. S1c, we plot the variation of the Stokes parameters across different sampling positions (e.g., pixels) within the region of interest. Hence, the horizontal axis does not represent a physical distance but rather an index of sampled points. To avoid confusion, we clarify this point in the Fig. S1c caption, stating that these ticks indicate sampling positions (pixel indices), not a physical length scale: “(c) (i) Variation in each Stokes parameter along the red line in (c), before and after SLM and DM calibration. Here, the horizontal axis represents sampling positions (pixel indices). (ii) Calculated precision along the same line and uniformity values of the whole plane before and after SLM and DM calibration (see Method 2).”

2) For the definition of precision and uniformity, we added the detailed descriptions into SI Note 1 for clarify:

“To evaluate the performance of calibration procedure, we introduce two criteria: vectorial precision and vectorial uniformity. We first quantify the vector difference D on the Poincaré sphere between the target SoP $S = [S_1, S_2, S_3]$ and the experimentally obtained SoP $\tilde{S} = [\tilde{S}_1, \tilde{S}_2, \tilde{S}_3]$ as:

$$D = c_p \|S - \tilde{S}\|$$

where both S and \tilde{S} are unit vectors ($[-1,1]$ range for each component) and c_p is a constant normalization factor chosen to keep D within $[0,1]$. Since $\|S - \tilde{S}\|$ can vary within the range of $[0,2]$, we set $c_p = \frac{1}{2}$.

We then define the precision metric P :

$$P = 1 - D$$

such that $P = 1$ signifies perfect matching between the target and actual SoP, while $P = 0$ represents the maximum mismatch on the Poincaré sphere.

The uniformity difference U_D is defined as the standard deviation of the actual SoP (\tilde{S}) from the mean SoP (\bar{S}) across the whole region of interest:

$$U_D = c_u \sqrt{\|\bar{S} - \tilde{S}\|^2}$$

where c_u is the constant normalization factor set to 1 to ensure U_D falls within $[0,1]$. A smaller U_D represents a more uniform distribution. Therefore, we define the uniformity metric U so that larger values of U indicate more uniform SoP distribution:

$$U = 1 - U_D''$$

Comment 4: *Application 1: How were the spatial distributions of the retarders calculated in more general terms? Figure 2a: How were the experimental maps retrieved from the measurements made by the Stokes polarimeter? Figure 2b: How were the 9 (experimental) Mueller matrices measured?*

Response 4: We thank the reviewer for the insightful questions regarding the calculation and measurement processes.

1) In Application 1, the spatial retarder distributions were calculated by solving the inverse problem between the known input vectorial beam and the target vectorial beam, as described in the Main article (as well as in Method 1). The expected output Stokes parameters were computed based on the target beam characteristics, and the corresponding pixel-wise retardance values were first obtained and then encoded onto the SLMs using a calibration-based look-up table (see Method 2). This ensures that the calculated distributions accurately reproduce the target beam. Then the full beam profile was then measured and verified using a Stokes polarimeter.

We note that different strategies can be used to calculate the phase profiles on the SLMs when generating complex beams with three SLMs. For uniform input and non-uniform output beam profiles, two-SLM-based methods¹¹ can provide a unique phase pattern solution (with retardance values from 0 to 2π and 0 to π) while keeping the third SLM flat. These unique patterns are then applied to the two SLMs, as demonstrated in most of the experimental results shown in Fig. 2a. For non-uniform input and output profiles, all three SLMs must be used, but the phase patterns solutions are non-unique. In this case, criteria such as minimal retardance should be applied to select the optimal phase solution. Notably, the three-SLM strategy can also be applied to uniform input cases, as shown in Fig. 2b.

We have revised the Main article (Application 1, paragraph 2) to highlight the above points: “The procedure for beam generation is as follows: First, an arbitrary field with known polarisation and phase is incident upon the array (we initially use a beam with circular polarisation and a flat phase for simplicity). Second, we determine the expected output fields. Third, we calculate the corresponding spatial retarder distributions and encode them into the retarder array sequentially (see Methods 1 and 2). Fourth, we use a Stokes polarimeter to record the beam profiles¹².”

And we have also revised Method 1: “Building on this foundation, there are different strategies for calculating the phase profiles on the SLMs depending on the input and output beam profiles. For uniform input and non-uniform output beam profiles, two-SLM-based methods¹¹ can provide a unique phase pattern solution (with retardance values from 0 to 2π and 0 to π) while keeping the third SLM flat. These unique patterns are then applied to the two SLMs, as demonstrated in most of the experimental results shown in main article Fig. 2a. For non-uniform input and output profiles, all three SLMs must be used, but the phase patterns solutions are non-unique. In this case, criteria such as minimal retardance should be applied to select the optimal phase solution. Notably, the three-SLM strategy can also be applied to uniform input cases, as shown in Fig. 2b.”

2) To generate the experimental maps in Fig. 2a, we employed a Stokes polarimeter consisting of a quarter-wave plate (QWP) and a linear polariser. The steps are as follows: by rotating the QWP to four distinct angles, we captured intensity images of the entire beam using a CCD camera. Each pixel in these images corresponds to a specific point on the beam, allowing us to measure the state of polarisation (SoP) at each location—this is a widely used Stokes vector measurement procedure, as properly cited in Ref [25] in the Main article.

For visualisation purposes, the polarisation ellipses were plotted from down-sampled data using a 10×10 averaging kernel at 10-pixel intervals, applied to the measured data of size 300×300 . This approach balances spatial resolution and visual clarity. Each ellipse represents the local SoP, with the major axis indicating the degree of linear polarisation, the ellipticity indicating the degree of circular polarisation, and the colour encoding the last component of Stokes parameter. As a same action as we have done in previous Response 1, we have added extra explanations in caption of Fig. S1b in SI Note 1 to address this point: “(b) ... To enhance visual clarity, the phase profile on the DM is illustrated and the Stokes ellipses are plotted based on down-sampled measurement data averaged over neighbouring 10×10 pixels.”

3) For Fig. 2b, the dual rotating waveplate polarimetry method is used to measure the experimental Mueller matrices¹³. In this method, two QWPs rotate at harmonic rates, with the second QWP rotating five times faster than the first. Two fixed polarisers are placed before the first QWP and after the second QWP. The arbitrary retarder array is positioned between the two QWPs, introducing the polarisation transformation being measured. The detected intensity modulation encodes this transformation, and Fourier analysis extracts the harmonic components linked to the 16 elements of the 4×4 Mueller matrix. Since the array functions as a pure retarder, only the 3×3 submatrix describing retardance is relevant and presented in the figure. Details can be found in Ref¹³, which we have properly cited in Main article as Ref [26].

11. Dai, Y. et al. Active compensation of extrinsic polarization errors using adaptive optics. *Opt Express* 27, 35797–35810 (2019).

12. Azzam, R. M. A. Stokes-vector and Mueller-matrix polarimetry. *JOSA A* 33, 1396–1408 (2016).

13. Goldstein, D. H. Mueller matrix dual-rotating retarder polarimeter. *Appl Opt* 31, 6676–6683 (1992).

Comment 5: *Application 2: Figure 3a: The colored bars of the polarimetric images are not clear, as well as the y-axis of the statistical comparison for the intensities and the polarimetric parameter. A double y-axis could clarify the plot. Regarding the Mueller matrices: how does the polarimetric parameter depend on the number of repeated units (if any) on the SLMs? Figure S3d: It would be interesting to see the two Cartesian and Polar optimization methods applied to the two samples. What are the Stokes parameters measured with the polarimeter (ground truth) compared to those obtained with the proposed methods? Can the authors comment on the advantages of using this method compared to the one reported in Ref. 25?*

Response 5: Many thanks for the valuable comments. Please let us address them one by one below:

1) For Fig. 3a, an HSL-based colour map is used for the polarimetric images. The hue (H) represents the fast axis orientation, defined by the angle between the Stokes parameter S_1 and S_2 . The lightness (L) reflects the retardance of the measured area. The saturation (S) is the constant 1. This encoding provides an intuitive and accurate visual representation of the polarimetric parameters. The captions for Fig. 3 and Fig. S3 have been revised to clarify this encoding: "... In the HSL colour map used, the hue (H) represents the fast axis orientation; the lightness (L) reflects the retardance of the measured area; and the saturation (S) is the constant 1." Regarding the bar plot, the values for intensity, ground truth, and polarimetric parameters are normalised to [0, 1] to enable direct comparison across different quantities. The y-axis represents the normalised values of these parameters. We have added axis descriptions in the Fig. 3, and the caption of Fig. 3 has also been revised accordingly: "... The statistical comparison of normalized values of the intensity, decomposed polarimetric parameter, and ground truth taken by traditional polarisation microscope are given, with data obtained from Z1 and Z2. ...".

2) Regarding the Mueller matrices, no repeated units are used on the SLMs, and each pixel is independently programmed to encode the local vectorial modulation. The measured Mueller matrix reflects the intrinsic local polarimetric response, determined solely by the modulation at each pixel and unaffected by pixel count or spatial repetition. This ensures that the overall polarimetric behaviour remains consistent across the entire modulation pattern.

3) Following the reviewer's comments, we have conducted additional experiments using traditional Stokes polarimeter to calculate the ground truth (GT) Stokes vectors for both samples from Zone 1 and Zone 2. Table S1 is provided, along with revised content, in SI Note 3 as follows: "... Additional statistical analyses are presented in Figure S3d, which shows the distribution and variance of the measured parameters, and Table S1, which compares the Stokes parameters (S_1 , S_2 , S_3) obtained using our proposed methods against those measured by a standard Stokes polarimeter (serving as ground truth, GT). To ensure a fair statistical comparison, we use 15 ROIs as an example. The results show approximately ~1% variation between the proposed method (ROI 15) and the GT for both biomedical and archaeological samples at two different zones (Zone 1 and Zone 2), confirming its high precision and reliability. ...

Table S1: Comparison of Stokes parameters from the proposed method and GT.

	Biomedical sample					
	Zone 1			Zone 2		
	S1	S2	S3	S1	S2	S3
ROI 15	-0.2631	0.2032	0.4078	-0.0201	0.0231	0.8372
GT	-0.2648	0.2092	0.3970	-0.0275	0.0173	0.8257
	Archaeological sample					
	Zone 1			Zone 2		
	S1	S2	S3	S1	S2	S3
ROI 15	0.4069	-0.7652	-0.0457	-0.2637	-0.5978	-0.6011
GT	0.3945	-0.7502	-0.0492	-0.2509	-0.6069	-0.5903

”

From further analysis, we find that our new approach exhibits approximately ~1% variation compared to traditional Stokes polarimeter results in S_1 , S_2 , and S_3 , validating the feasibility and high precision of our method. Note that we use 15 ROIs as an example to ensure a fair comparison from a statistical point of view. Regarding the comparison between CO and PO for different samples, please allow us to leave this for future work. The reason is that this requirement could actually lead to a separate future paper, while in the present study, we aim for this application to serve as a proof-of-concept for the arbitrary retarder array, primarily focusing on the new research directions enabled by these elliptical retarders, particularly in a tuneable format. So, the current form may already serve the purpose. Additionally, please allow us to quote comments of Reviewers 1 and 2, who previously mentioned that the paper already covers too many different directions in detail (which is why we have already deleted some content). Therefore, we hope to aim to keep it as concise as possible. However, we greatly appreciate your comments, so we have added new experimental results in Table S1, as mentioned earlier. We hope you kindly agree and support this action.

4) Thanks for the very good question -- Ref. [25] is based on a polarisation camera with circular SoP illumination to calculate retardance and fast-axis orientation from the Stokes vectors with respect to a thin sample. Essentially, the setup is passive and not tuneable, and it is also restricted by the traditional measurement paradigm (matrix calculation), which is limited by error amplification in matrix calculations. In contrast, our method not only employs a new paradigm (see Ref.¹⁴ for details) but also first time makes the system tuneable. As we explained before, the main aim of this work is to optimise the overall polarisation sensing system into a novel, target-optimised state -- in which tunability is a crucial capability, making our work significantly different from previous approaches. We have emphasised this again in SI Note 3 for further clarification: “The tunability of the analysing channel is important, as different applications require different optimisation strategies for SoP sensing. For instance, in certain pathological imaging scenarios, the measurement precision of circular SoP is more critical than its linear counterpart¹⁵, whereas linear SoP is given more consideration in certain material characterisation applications¹⁶...”

14. He, C. et al. Full Poincaré polarimetry enabled through physical inference. *Optica* 9, 1109–1114 (2022).

15. Chang, J. et al. Division of focal plane polarimeter-based 3×4 Mueller matrix microscope: a potential tool for quick diagnosis of human carcinoma tissues. *J Biomed Opt* 21, 56002 (2016).

16. Cui, M., Guan, M., Zhu, J., Hu, R. & Zhu, J. *Research on bluish-white porcelain glazes of Fanchang kiln in China. Archaeometry* (2024).

Comment 6: *How many iterations were performed in the O-AO (sensor and sensor-less) approach? How much does the convergence of this process depend on the deviation of the input beam from the desired output? Are there any limitations?*

Response 6: We appreciate the reviewer's insightful question about the O-AO approach.

1) In the sensor-based O-AO approach, no iteration is required since the aberration is measured directly, and the complementary correction can be calculated accordingly (please see details in Method 5 and SI Note 4). For the sensorless method, 20 vectorial retardance modes (Z_1 to Z_{20} ; please see details in SI Note 5) are iterated, with each mode coefficient scanned from -1.0 to 1.0 in steps of 0.1 (also please see details in Method 5 and SI Note 5). This results in 420 measurements for a full sensorless correction cycle. Method 5 has been revised to address this point: "... By utilizing Mueller matrix polarimetry, the polarisation aberration D introduced by the external sample can be quantitatively obtained without any iterations. ..." and "... Before the recording of each mode pattern, the optimal α_k needs to be determined referring to FID images by interating it from -1.0 to 1.0 in steps of 0.1. ..."

2) For small deviations, the initial few mode scans (less than 10) are usually sufficient to correct the aberration, whereas larger deviations require iterations of more modes to achieve convergence.

3) The main limitations arise when the deviation is complex and exceeds the available mode range, which increases the number of iterations required and reduces the overall correction accuracy. Additionally, the correction quality may degrade under strong superposition of the mode patterns due to the SLM pixel resolution, particularly for large deviations or high-frequency aberrations. However, these limitations are highly likely to be mitigated by refining the calibration process for mode superposition with more modes using higher-resolution SLMs. Moreover, enhanced learning-based correction algorithms could further improve accuracy and efficiency. We have revised Method 5 to highlight this point: "It is worth noting that the correction accuracy may be reduced when the deviation is complex or exceeds the available mode range, leading to slower convergence and increased iterations. Additionally, the correction quality may degrade under strong superposition of mode patterns due to the SLM pixel resolution, particularly for large deviations or high-frequency aberrations. However, these limitations are highly likely to be mitigated by refining the calibration process for mode superposition with more modes using higher-resolution SLMs. Furthermore, expanding the mode range with enhanced learning-based correction algorithms¹⁷ could further enhance accuracy and efficiency."

17. Hu, Q. et al. *Universal adaptive optics for microscopy through embedded neural network control. Light Sci Appl* 12, 270 (2023).

We thank very much again for all of the Reviewer's valuable comments, and hope the revised version now meets the reviewer's expectations.

Reply to Reviewer #3

Comment: *I appreciate the Authors' detailed responses and revisions to the text. The manuscript now includes more experimental details, better explains the results obtained and the procedure used. This makes the work clearer and more reproducible for the wide readership of Nature Communication. I recommend publication of the article.*

Response: We sincerely thank the reviewer for the encouraging feedback and support for publication. Thank you again for all the constructive input throughout the revision process.